environmental chemistry

chemical mixtures, endocrine disruptor chemicals, semivolatile organic compounds, flame retardants, phthalates, exposure science

**Author for correspondence:**
Kim A. Anderson
e-mail: kim.anderson@oregonstate.edu

This article has been edited by the Royal Society of Chemistry, including the commissioning, peer review process and editorial aspects up to the point of acceptance.

# Discovery of common chemical exposures across three continents using silicone wristbands

Holly M. Dixon[1], Georgina Armstrong[5], Michael Barton[1], Alan J. Bergmann[1], Melissa Bondy[5], Mary L. Halbleib[2], Winifred Hamilton[6], Erin Haynes[9], Julie Herbstman[10], Peter Hoffman[1], Paul Jepson[3], Molly L. Kile[4], Laurel Kincl[4], Paul J. Laurienti[11], Paula North[12], L. Blair Paulik[1], Joe Petrosino[7], Gary L. Points III[1], Carolyn M. Poutasse[1], Diana Rohlman[4], Richard P. Scott[1], Brian Smith[1], Lane G. Tidwell[1], Cheryl Walker[8], Katrina M. Waters[13] and Kim A. Anderson[1]

[1]Food Safety and Environmental Stewardship Program, Environmental and Molecular Toxicology, [2]Department of Crop and Soil Science, [3]Integrated Plant Protection Center, and [4]College of Public Health and Human Sciences, Oregon State University, Corvallis, OR, USA
[5]Department of Medicine, Section of Epidemiology and Population Sciences, [6]Department of Medicine, Environmental Health Section, [7]Department of Molecular Virology and Microbiology, and [8]Department of Medicine, Center for Precision Environmental Health, Baylor College of Medicine, Houston, TX, USA
[9]College of Medicine, Department of Environmental Health, University of Cincinnati, Cincinnati, OH, USA
[10]Columbia Center for Children's Environmental Health, Department of Environmental Health Sciences, Mailman School of Public Health, Columbia University, New York, NY, USA
[11]Department of Radiology, Wake Forest School of Medicine, Winston-Salem, NC, USA
[12]Department of Pathology, Medical College of Wisconsin, Milwaukee, WI, USA
[13]Biological Sciences Division, Pacific Northwest National Laboratory, Richland, WA, USA

HMD, 0000-0003-0326-4477; CMP, 0000-0001-7128-044X;
KMW, 0000-0003-4696-5396; KAA, 0000-0002-5258-2925

To assess differences and trends in personal chemical exposure, volunteers from 14 communities in Africa (Senegal, South Africa), North America (United States (U.S.)) and South America (Peru) wore 262 silicone wristbands. We analysed wristband extracts for 1530 unique chemicals, resulting in

400 860 chemical data points. The number of chemical detections ranged from 4 to 43 per wristband, with 191 different chemicals detected, and 1339 chemicals were not detected in any wristband. No two wristbands had identical chemical detections. We detected 13 potential endocrine disrupting chemicals in over 50% of all wristbands and found 36 chemicals in common between chemicals detected in three geographical wristband groups (Africa, North America and South America). U.S. children (less than or equal to 11 years) had the highest percentage of flame retardant detections compared with all other participants. Wristbands worn in Texas post-Hurricane Harvey had the highest mean number of chemical detections (28) compared with other study locations (10–25). Consumer product-related chemicals and phthalates were a high percentage of chemical detections across all study locations (36–53% and 18–42%, respectively). Chemical exposures varied among individuals; however, many individuals were exposed to similar chemical mixtures. Our exploratory investigation uncovered personal chemical exposure trends that can help prioritize certain mixtures and chemical classes for future studies.

# 1. Introduction

## 1.1. Personal exposure to chemical mixtures

People are exposed to complex chemical mixtures, rather than to a single chemical or an individual chemical class [1–3]. Yet, toxicological and epidemiological studies often focus on one chemical or chemical class. Chemical mixtures may result in significantly different toxicities compared with individual chemical components, with the potential for additive, synergistic or antagonistic effects [1]. In certain circumstances, assessing health risks by individual chemicals may underestimate actual risks because the combined effect of two chemicals is greater than the sum of both independent effects [2]. To better understand the link between real-world chemical exposure and health effects, simple personal monitoring methods are necessary to capture exposures from multiple chemical classes [3,4].

Certain chemical exposures are associated with adverse health outcomes. Briefly, exposure to certain polycyclic aromatic hydrocarbons (PAHs) has been associated with cancer [5], self-regulatory capacity issues [6], low birth weight [7] and respiratory distress [8]. Exposure to specific flame retardants has been associated with cancer [9,10], neurotoxicity [9,11] and cardiotoxicity [12]. Exposure to endocrine-disrupting chemicals (EDCs) has been linked to health effects such as low semen quality, adverse pregnancy outcomes and endocrine-related cancers [13,14]. Simultaneously assessing personal exposure to multiple chemical classes, such as PAHs, flame retardants, pesticides, phthalates and EDCs, may help researchers better connect chemical exposure to health.

There is a need to characterize common EDC mixtures because EDCs span several chemical classes and have the potential to create significant health effects by altering hormone activities. The 2017 National Academies of Science Endocrine Report proposed that the EPA develops a low-dose EDC exposure surveillance programme, which would include the collection of personal exposure data [15]. A 2015 research study on EDCs highlights the lack of personal exposure information for EDC mixtures [16]. A 2012 report from the World Health Organization states that 'more comprehensive assessments of…exposures to diverse mixtures of EDCs are needed' [17]. To assess personal exposure to low-dose EDCs, the 2017 National Academies Report recommended that researchers use external chemical exposure data in the absence of biomonitoring data, and suggested that silicone wristbands could be used to assess individuals' chemical exposure [15]. Silicone wristbands are a novel application of passive sampling that O'Connell et al. first described in 2014 [18]. Wristbands have been identified as an 'unprecedented measurement platform' that provides individualized chemical exposure data [19], which can be applied to many different types of environmental epidemiological studies [20].

## 1.2. Wristbands to assess personal exposure to chemical mixtures

In this study, we use silicone wristbands to assess personal exposure to chemical mixtures. Since 2014, researchers have deployed wristbands around the world to evaluate personal chemical exposure. To date, researchers have used wristbands to detect chemical exposure in communities ranging from preschoolers in Oregon, U.S. to farming families in Senegal, Africa, and wristband results are described in 14 peer-reviewed manuscripts [18,21–33].

Although O'Connell *et al.* first reported using wristbands for passive sampling in 2014, researchers have used passive sampling methods for over two decades to sample the bioavailable fraction of organic chemicals in air, sediment and water [34–38]. Unbound volatile organic compounds (VOCs) and semivolatile organic compounds (SVOCs) in the environment diffuse into the lipophilic membrane of passive sampling polymers [24,34], which results in passive samplers reflecting the bioavailable fraction of chemical exposure [35,39–41].

Wristbands have been compared with other personal exposure assessment methodologies, including organophosphate flame retardants (OPFRs) in hand wipes and urine [23] and PAHs in air monitoring backpacks and urine [29]. In these two studies, chemical concentrations in wristbands and paired metabolites in urine samples were highly correlated, providing further evidence that wristbands are a biologically relevant surrogate of chemical exposure [23,29].

Because the silicone polymer sequesters a broad range of organic chemicals [14], wristbands are especially suitable for analysis of several chemical classes concurrently. The wristband's ability to capture and retain VOCs and SVOCs is detailed in Anderson *et al.* [22]. Researchers can easily archive wristbands and/or wristband extracts to re-analyse for additional chemicals as analysis methods expand and additional study questions arise.

Wristbands detect chemicals in an individual's external environment, incorporating both inhalation and dermal exposure [29]. This is an asset for emerging research focused on the exposome and measuring the totality of personal exposure to chemicals [42,43]. Wristbands sample chemicals in the gaseous phase and many SVOCs exist in this phase in concentrations relevant to human health. Both low- and high-molecular-weight PAHs have been found in the gaseous phase [7,44–46] and can be major contributors to PAH-associated adverse health effects [47]. Polybrominated diphenyl ethers (PBDEs) and OPFRs in the gaseous phase have been reported to be just as critical to assess for inhalation and dermal exposure as when associated with a particulate matter [10,48,49].

## 1.3. Study objectives

We used silicone wristbands collected in 14 unique communities and analysed 262 wristband extracts for the presence–absence of 1530 organic chemicals. The objectives of this work were: (i) to demonstrate the use of wristbands as a screening tool for population exposures to organic chemicals, (ii) to investigate individual and community exposures to 1530 chemicals including over 400 potential EDCs and (iii) to compare chemical detections between various demographic and geographical variables. We hypothesized that comparing chemical detections between different communities would reveal chemical exposure patterns that could inform future toxicology and epidemiology research.

# 2. Material and methods

## 2.1. Study participants and design

To represent as many chemical exposures as possible, this exploratory, retrospective study includes 262 wristbands worn by 246 volunteers on three continents from multiple prior wristband studies (table 1). The research ethics section of this paper includes details on all Institutional Review Board (IRB) approval and informed consent. Volunteer gender, age, population density and community for each wristband are included in the electronic supplementary material. Volunteers wore the wristband for the entire study period and were asked not to alter their daily activities.

## 2.2. Wristband methodology

### 2.2.1. Preparation and deployment

We purchased silicone wristbands from 24hourwristbands.com (Houston, TX, USA). We initially rinsed wristbands with deionized water to remove potential surface particulates and then conditioned wristbands to remove chemicals of interest from the silicone which is described in O'Connell *et al.* [18], Donald *et al.* [25] and Anderson *et al.* [22]. Prepared wristbands were stored in airtight metal containers at 4°C. For deployment, wristbands were individually packaged in airtight polytetrafluoroethylene (PTFE) bags (Welch Fluorocarbon, Dover, NH, USA) and labelled according to study protocols.

**Table 1.** Description of the different geographical and demographic variables associated with the 262 wristbands in this study.

| | Africa | | North America | | | | | | | South America | totals |
|---|---|---|---|---|---|---|---|---|---|---|---|
| community | 1 | 2 | 3–4 | 5 | 6 | 7 | 8 | 9 | 10 | 11–14 | |
| country | Senegal | South Africa | U.S. | U.S. | U.S. | U.S. | U.S. | U.S. | U.S. | Peru | |
| region | Diender | Cape Town | Oregon *Corvallis and Bend* | New York City | Ohio *Carroll County area* | Washington, DC | Oregon *Eugene* | North Carolina *Winston-Salem & Benson areas* | Texas[b] *Houston area* | Alto Mayo | |
| number of volunteers | 25 | 2 | 21 | 22 | 24 | 24 | 11 | 22 | 26 | 69 | 246 |
| number of wristbands | 25 | 2 | 21 | 24 | 24 | 24 | 25 | 22 | 26 | 69 | 262 |
| time of study | Nov.–Dec. 2014 | spring 2015 | Oct. 2012–Jan. 2013 | 2013–2015 | May–June 2014 | May 2015 | 2013 & 2015 | July–Aug. 2016 | Sept. 2017 | Feb.–Mar. 2014 | |
| related reference | Donald et al. [25] | — | Kile et al. [24] | Dixon et al. [29] | Paulik et al. [30] | — | — | Vidi et al. [26] | — | Bergmann et al. [31] | |
| population density[a] rural | 25, 100% | — | — | — | 24, 100% | — | — | 10, 45% | — | 49, 71% | 108, 41% |
| urban | — | 2, 100% | 21, 100% | 24, 100% | — | 24, 100% | 25, 100% | 12, 55% | 26, 100% | 20, 29% | 154, 59% |
| age[a] < 11 years | — | — | 21, 100% | — | — | — | — | 20, 91% | — | 4, 6% | 45, 17% |
| 11–20 years | 1, 4% | — | — | — | — | — | — | — | — | 9, 13% | 10, 4% |
| 21–40 years | 16, 64% | — | — | 23, 96% | — | — | 3, 12% | — | 7, 27% | 22, 32% | 71, 27% |
| 41–60 years | 4, 16% | — | — | 1, 4% | — | — | 4, 16% | — | 10, 38% | 24, 35% | 43, 16% |
| >60 years | — | — | — | — | — | — | 2, 8% | — | 5, 19% | 9, 13% | 16, 6% |
| not available | 4, 16% | 2, 100% | — | — | 24, 100% | 24, 100% | 16, 64% | 2, 9% | 4, 15% | 1, 1% | 77, 29% |
| gender[a] female | 3, 12% | 1, 50% | — | 24, 100% | 14, 58% | — | 8, 32% | 10, 45% | 12, 46% | 35, 51% | 107, 41% |
| male | 22, 88% | — | — | — | 9, 38% | — | 17, 68% | 12, 55% | 10, 38% | 33, 48% | 103, 39% |
| not available | — | 1, 50% | 21, 100% | — | 1, 4% | 24, 100% | — | — | 4, 15% | 1, 1% | 52, 20% |

[a]We report the number of wristbands (and percentage of total wristbands) associated with each variable within each community. Owing to rounding, not all percentages add up to 100.
[b]Wristbands in Texas were deployed within a month of Hurricane Harvey making landfall.

Before deployment, we collected blank wristbands from each group of conditioned wristbands and we analysed the blank wristbands using gas chromatography–mass spectrometry (GC–MS) with perylene-d12 (500 ng) as an internal standard. To ensure the removal of oligomers that can adversely impact analytical sensitivity, we verified that (i) there were less than four discrete peaks over 15 times the response of our internal standard in the total ion chromatogram of each extract and (ii) that the total mass reduction of conditioned wristbands compared with pre-conditioned wristbands was greater than 3%. To reduce unnecessary sample loss, we also verified that wristbands retained elasticity. Wristbands were not deployed unless these criteria were met.

### 2.2.2. Cleaning and extraction

After deployment, wristbands were returned in airtight PTFE bags to Oregon State University (OSU) for analysis. To remove surface fouling and particulates, deployed wristbands were cleaned twice with 18 MΩ cm water and once with isopropanol, and then stored in amber jars at $-20°C$ until extraction [18,21,22,25].

All 262 wristbands were solvent extracted as reported in O'Connell et al. [18]. We added extraction surrogates and then extracted chemicals from wristbands with two rounds of ethyl acetate (100 ml) at room temperature. We quantitatively concentrated the ethyl acetate using TurboVap® evaporators (Biotage LLC, Charlotte, NC, USA). Wristband extracts were stored at $-20°C$ until analysis. For 35% of wristbands, we conducted solid-phase extraction (SPE) after solvent extraction. For SPE, we added 3 ml of acetonitrile to each sample, which were then loaded onto pre-rinsed C18 SPE cartridges at $1.8 \, ml \, min^{-1}$ (Supelco, Bellefonte, PA, USA; O'Connell et al. [18] and Kile et al. [24]). Samples were eluted at $3 \, ml \, min^{-1}$ with 9 ml of acetonitrile (Rapid Trace, automated SPE workstation, Biotage, Uppsala, Sweden) [18,24]. SPE further cleans samples containing high levels of fats (e.g. fatty acid esters and chains) and/or oils in personal care products that might interfere with chemical analyses.

### 2.2.3. Chemical analysis

We used an Agilent 7890A GC interfaced with an Agilent 5975C MS detector to analyse all wristband extracts for the presence–absence of 1530 organic chemicals (GC–MS control parameters listed in electronic supplementary material, table S1; Bergmann et al. [31]). An Agilent DB-5MS column (30 m × 0.25 mm) was used in the GC and the inlet pressure was retention-time locked to chlorpyrifos [31]. This high-throughput screen uses an automated mass spectral deconvolution and identification system (AMDIS v. 2.66, National Institute of Standards and Technology) paired with deconvolution reporting software (DRS, Agilent) to identify the presence–absence of 1530 chemicals. These chemicals were selected because they may influence human health. This target list includes 76 consumer product-related chemicals, 124 flame retardants, 185 industrial-related chemicals, 98 PAHs, 260 PCBs/dioxins/furans, 773 pesticides and 14 phthalates. A list of target chemicals is available at http://fses.oregonstate.edu/1530. Bergmann et al. [31] report limits of quantitation for all 1530 chemicals in the analytical method used in this paper, which range from 40 to 500 pg $\mu l^{-1}$ depending on the chemical.

We manually reviewed chemicals with a greater than or equal to 60% match to library spectra in a process which protects against false positives. We evaluated each individual chromatogram processed with AMDIS according to our data quality objectives (DQOs) [31]. If the following criteria were not met, the peak was excluded from our analysis: retention time shifts must be less than 45 s, peak responses must be greater than a 3 : 1 signal-to-noise ratio and the peak shape of AMDIS extracted ions must match the sample's extracted peak shape. Chemists also looked for missing or extra $m/z$ peaks on each extracted spectrum in comparison with the corresponding AMDIS library spectrum, which can also lead to peak exclusion [31].

### 2.2.4. Quality control summary

Quality control (QC) steps were included to ensure data quality. We collected and analysed blank wristbands that travelled to and from study locations. We also analysed solvent that went through the entire extraction process without a wristband. We analysed instrument blanks and calibration verifications (CVs) every 10–15 samples. To meet our DQOs, all target chemicals were below the instrument detection limits in the ethyl acetate or hexane instrument blanks. Prior to instrumental analysis, we positively identified greater than 80% of target chemicals in the CVs. If a CV did not meet our DQOs, we verified our standards and, if needed, performed instrument maintenance before

re-running samples. Our series of QC throughout wristband conditioning, travelling, cleaning, extraction and analytical processes allowed us to account for any potential chemical contamination. For more QC results on specific projects, we refer to the related wristband manuscripts [21,24–26,29,30].

We have analysed several hundred blank wristbands and found all target chemicals below detection limits with the exception of a few phthalates. Blank wristbands can contain some of the 1530 target chemicals and pass our DQOs if the amounts in blank wristbands are at least 100 times lower in concentration than deployed wristbands. The phthalates we regularly identify in blank wristbands are typically 100–10 000 times lower in concentration than deployed wristbands, which we track and monitor during the chemical analysis process [31].

### 2.2.5. Chemicals and solvents

We purchased chemical standards from Accustandard (New Haven, CT, USA), Sigma-Aldrich (St. Louis, MO, USA), TCI America (Portland, OR, USA), Santa Cruz Biotechnology (Dallas, TX, USA) and Chiron (Trondheim, Norway). All solvents were Optima-grade or equivalent (Fisher Scientific, Pittsburgh, PA, USA). All of the tools and glassware were baked for 12 h at 450°C and/or solvent-rinsed before use. For processes requiring 18 MΩ cm water, the water was filtered through a D7389 purifier (Barnstead International, Dubuque, IA, USA).

## 2.3. Data analysis

We assigned each community an urban or rural classification. An urban classification includes both urban and suburban communities—settlements with medium to high population density. A rural classification includes areas with low population density and small settlements. We acknowledge that there can be large differences in the human-made surroundings between rural (or urban) communities depending on the country and socio-economic class.

We categorized participant age into five groups: under 11, 11–20, 21–40, 41–60 and over 60. These age groups are similar to what is used by the U.S. National Health and Nutrition Examination Survey (NHANES) [50], although we further divided the NHANES 20–59-year-old group into two groups to look for additional chemical detection patterns.

Each chemical was assigned one of seven primary categories: (i) consumer product-related chemicals; (ii) flame retardants; (iii) industrial-related chemicals; (iv) PAHs; (v) PCBs, dioxins and furans; (vi) pesticides and (vii) phthalates. We acknowledge most chemicals fit in more than one category. For example, triphenyl phosphate (TPP) is not only a flame retardant but also an industrial-related chemical (used as a plasticizer). For this study, we assigned TPP to the flame retardant category. Potential EDCs were categorized according to the Endocrine Disruptor Exchange List (https://endocrinedisruption.org/interactive-tools/tedx-list-of-potential-endocrine-disruptors/search-the-tedx-list; accessed September 2017). There are 432 potential EDCs in our analytical method.

We used sigma.js software to conduct a network analysis on co-occurring chemical detections. The outline of the proportional Venn diagrams for three sets was created using eulerAPE software as developed by Micallef & Rodgers [51]. Two wristband groups in the Venn diagrams for the 30% most commonly detected chemicals included one additional chemical in the Venn to account for ties in chemical detection numbers. We created boxplots and tree maps with JMP Pro, v. 13.2.1. We used Tukey–Kramer honestly significant difference (HSD) tests to compare differences between all possible pair of means for more than two wristband groups and Student's t-tests to compare differences between means for two groups. Statistical significance was set at $\alpha = 0.05$ for all analyses. South Africa wristband data ($n = 2$) were excluded from Tukey–Kramer HSD tests on regional and geographical density differences. The 11–20 age bin in Africa ($n = 1$) was also excluded for tests on age bin differences. Each section of the tree maps (e.g. Ohio, Africa rural and South American female) includes the seven pre-determined chemical categories. The size of each coloured box reflects the percentage of chemical detections for that specific chemical category. Principal component analysis (PCA) was applied to the presence–absence data for the 191 unique chemicals detected in this study, with presence indicated numerically as 1 and absence as 0 (using Primer-E, v.6). Pearson correlations underlie this PCA. A given vector points in the direction of increasing density of 1s corresponding to the associated chemical. Principal component (PC) pairs ranging from PC1 to PC5 were considered, with PC1 versus PC2 displayed.

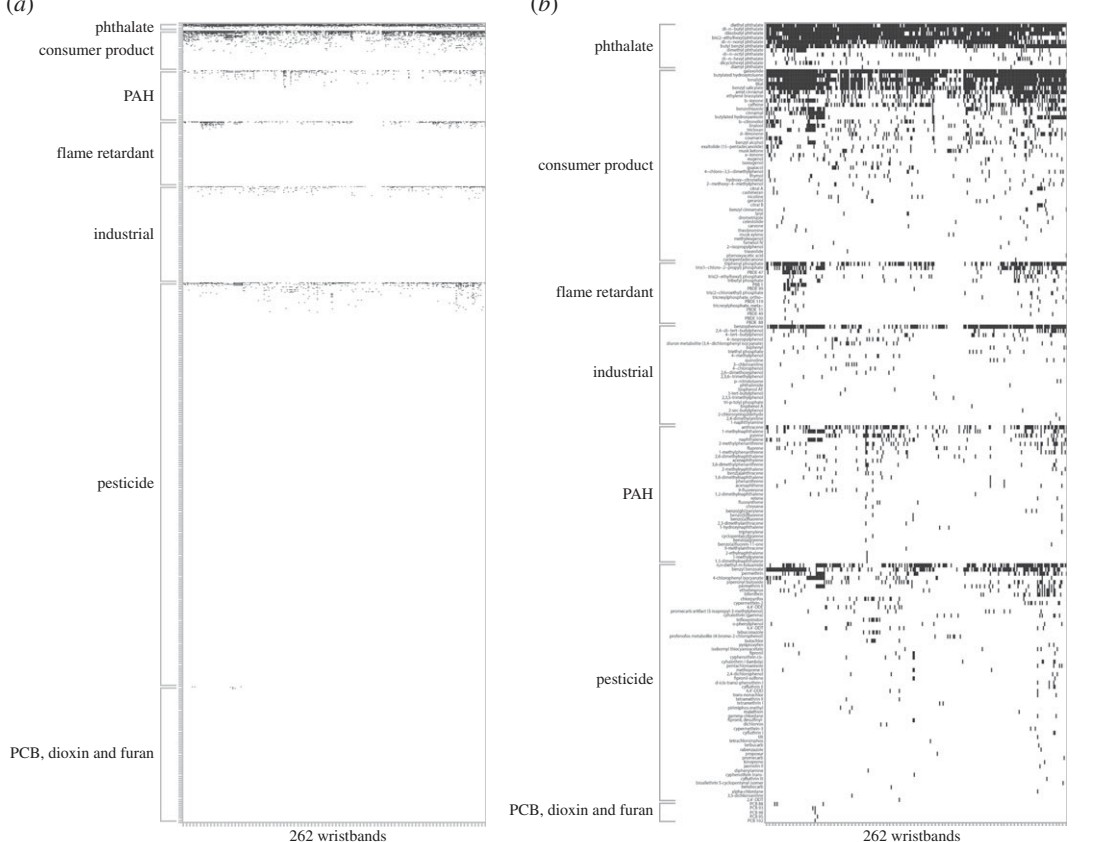

**Figure 1.** Heat map of (*a*) all 1530 organic chemicals tested for in the wristbands and of (*b*) all 191 chemicals detected at least once in the wristbands. Black indicates a chemical was detected in a wristband while white indicates a chemical was not detected.

# 3. Results

We analysed 262 wristbands from 246 volunteers. We have geographical information for all volunteers, including the country, the region within the country, and rural or urban designations. Because some volunteers wore more than one wristband, we will refer to groups of wristbands rather than groups of volunteers throughout the results section. We have participant-identified gender for 80% of wristbands and age for 71% of wristbands. Volunteer ages ranged between 3 and 86, with a median age of 34. The sample sizes for different demographic and geographical variables are summarized in table 1.

## 3.1. Wristband chemical detections

No two wristbands had the same chemical detection profile. Overall, chemical detections ranged from 4 to 43 per wristband, with an average of 20. One hundred and ninety-one chemicals were detected at least once. These chemicals, along with the additional 1339 chemicals that were not detected in any wristband, are visually represented in a heat map in figure 1. We detected 14 chemicals in over 50% of all wristbands, 13 of which are potential EDCs (table 2).

Detections of potential EDCs ranged from 4 to 30 per wristband, with an average of 14. Of the 191 chemicals we detected once, 96 are classified as potential EDCs (figure 2*a*,*b*) and 95 are not (figure 2*c*).

### 3.1.1. Venn diagrams

We detected 36 chemicals in common between wristbands worn in three continent-based wristband groups: (i) North America, (ii) South America and (iii) Africa (figure 3*a* and table 3). Of the 30% most commonly detected chemicals in wristbands worn in North America, South America and Africa, there were 13 chemicals in common (figure 3*b* and table 3). North American volunteers had (i) the highest number of chemicals detected compared with South America and Africa and (ii) the highest number

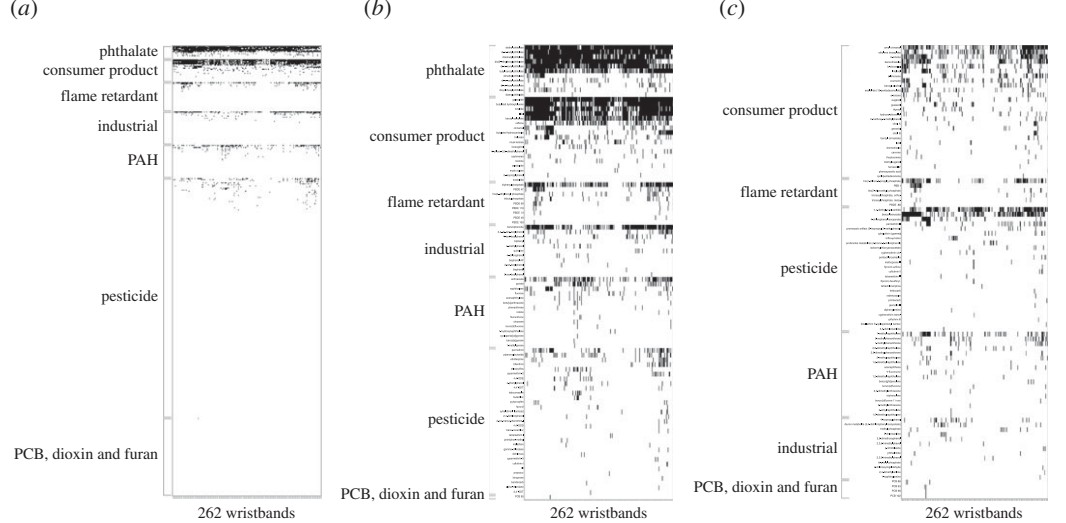

**Figure 2.** Heat map of (*a*) all 432 potential endocrine disrupting chemicals (EDCs) tested for in the wristbands, (*b*) all 96 potential EDCs detected at least once in the wristbands and (*c*) all 95 chemicals that are not potential EDCs and detected at least once in the wristbands. Black indicates a chemical was detected in a wristband while white indicates a chemical was not detected.

**Table 2.** Detection frequencies for chemicals found in greater than 50% of wristbands.

| chemical | frequency of detection out of 262 wristbands (%) | potential endocrine disruptor chemical | primary chemical category |
|---|---|---|---|
| diethyl phthalate | 95 | yes | phthalate |
| galaxolide | 94 | yes | consumer product-related |
| di-*n*-butyl phthalate | 93 | yes | phthalate |
| diisobutyl phthalate | 85 | yes | phthalate |
| bis(2-ethylhexyl)phthalate | 84 | yes | phthalate |
| di-*n*-nonyl phthalate | 82 | yes | phthalate |
| butylated hydroxytoluene | 79 | yes | consumer product-related |
| tonalide | 76 | yes | consumer product-related |
| lilial | 75 | yes | consumer product-related |
| benzyl salicylate | 73 | yes | consumer product-related |
| butyl benzyl phthalate | 66 | yes | phthalate |
| benzophenone | 64 | yes | industrial-related |
| triphenyl phosphate | 52 | yes | flame retardant |
| *n,n*-diethyl-*m*-toluamide | 52 | no | pesticide |

of chemicals detected unique to just one continent (figure 3, chemicals listed in electronic supplementary material, tables S2 and S3).

At the intersection of five population-density wristband groups: (i) U.S. urban, (ii) U.S. rural, (iii) Peru urban, (iv) Peru rural and (v) Senegal and South Africa, there were 28 chemicals in common (figure 3*c* and table 3). Twenty chemicals were found in common between four groups, 27 chemicals between three groups and 40 chemicals between two groups (chemicals listed in electronic supplementary material, table S4). When comparing the 30% most commonly detected chemicals between the five population-density groups mentioned above, there were 13 chemicals in common (figure 3*d* and table 3). Notably, these same 13 chemicals in common were at the intersection of the three other Venn diagrams (table 3). Of these 13 chemicals, 11 of them (benzophenone, benzyl salicylate, bis(2-ethylhexyl)phthalate, butylated hydroxytoluene, di-*n*-butyl phthalate, di-*n*-nonyl phthalate, diethyl

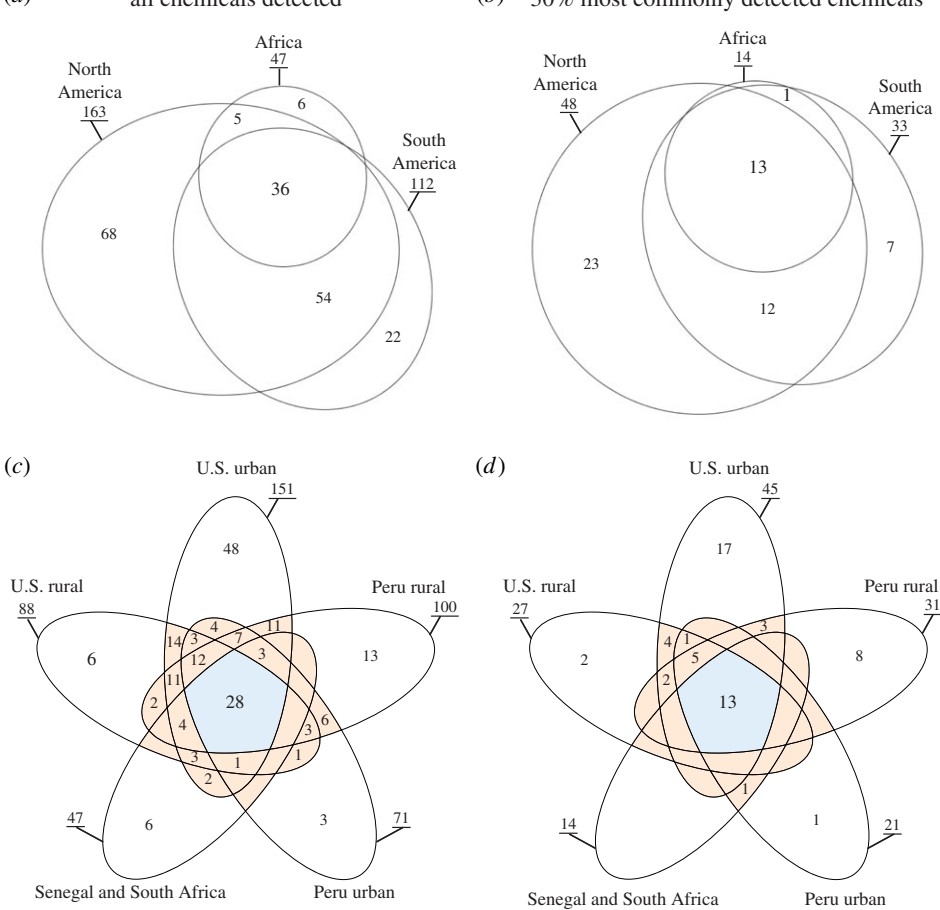

**Figure 3.** Commonalities between chemicals detected in North America, Africa and South America groups for (*a*) all chemicals detected in this study and for (*b*) the 30% most commonly detected chemicals. Commonalities between chemicals detected in U.S. rural, U.S. urban, Peru rural, Peru urban and Senegal & South Africa groups for (*c*) all chemicals detected in this study and for (*d*) the 30% most commonly detected chemicals. Venn diagrams (*a*) and (*b*) are area-proportional to the number of chemical detections at each intersection, which does not apply to the five-group Venn diagrams (*c*) and (*d*). Underlined numbers represent the total number of chemical detections found within each wristband group.

phthalate, diisobutyl phthalate, galaxolide, *n,n*-diethyl-*m*-toluamide and tonalide) were also detected in over 50% of all wristbands. In figure 4*d*, five chemicals were found in common between four groups, three chemicals were found in common between three groups and eight chemicals were found in common between two groups (chemicals listed in electronic supplementary material, table S5).

### 3.1.2. Network analysis

The same chemical pairs were often detected together in a wristband. Thirty-five different chemical pairs occurred in 20 or more wristbands in this study (electronic supplementary material, figure S1). Fourteen of those pairs include TPP, 10 include *b*-ionone and nine include benzothiazole. For example, TPP and *b*-ionone co-occurred in 65 wristbands. TPP and tris(1-chloro-2-propyl) phosphate (TCPP) co-occurred in 57 wristbands. Cinnamal and benzothiazole co-occurred in 31 wristbands.

## 3.2. Geographical and demographic differences

We examined differences in the number of chemicals detected and primary chemical categories for geographical and demographic variables (figure 4). The majority of chemicals detected were consumer product-related chemicals and phthalates, regardless of region, population density, age and gender. Specifics regarding chemical detection means and Tukey–Kramer HSD results are included in electronic supplementary material, tables S5–S8.

**Table 3.** Chemicals in common between three and five unique groups of wristbands, corresponding to figure 3. Chemicals in italics text indicate the chemical is listed in common for all four Venn diagrams.

| Venn diagram groups | North America<br>Africa<br>South America | | U.S. rural<br>U.S. urban<br>Peru rural<br>Peru urban<br>Senegal & South Africa | |
|---|---|---|---|---|
| detected chemicals included | all 191 chemicals<br>figure 3a | 30% most common<br>figure 3b | all 191 chemicals<br>figure 3c | 30% most common<br>figure 3d |
| number of chemicals in common between wristband groups | 36 | 13 | 28 | 13 |
| list of chemicals in common | *amyl cinnamal*<br>*benzophenone*<br>*benzyl salicylate*<br>*bis(2-ethylhexyl)phthalate*<br>*butylated hydroxytoluene*<br>*di-n-butyl phthalate*<br>*di-n-nonyl phthalate*<br>*diethyl phthalate*<br>*diisobutyl phthalate*<br>*ethylene brassylate*<br>*galaxolide*<br>*n,n-diethyl-m-toluamide*<br>*tonalide*<br>1-methylnaphthalene<br>2,6-dimethylnaphthalene<br>4-chlorophenyl isocyanate<br>anthracene<br>benzothiazole<br>benzyl benzoate<br>butyl benzyl phthalate<br>butylated hydroxyanisole<br>caffeine<br>coumarin<br>d-limonene<br>di-*n*-hexyl phthalate<br>di-*n*-octyl phthalate<br>eugenol<br>exaltolide (15-pentadecanolide)<br>fluorene<br>hydroxy-citronellal<br>lilial<br>linalool<br>musk ketone<br>pyrene<br>thymol<br>triclosan | *amyl cinnamal*<br>*benzophenone*<br>*benzyl salicylate*<br>*bis(2-ethylhexyl)phthalate*<br>*butylated hydroxytoluene*<br>*di-n-butyl phthalate*<br>*di-n-nonyl phthalate*<br>*diethyl phthalate*<br>*diisobutyl phthalate*<br>*ethylene brassylate*<br>*galaxolide*<br>*n,n-diethyl-m-toluamide*<br>*tonalide* | *amyl cinnamal*<br>*benzophenone*<br>*benzyl salicylate*<br>*bis(2-ethylhexyl)phthalate*<br>*butylated hydroxytoluene*<br>*di-n-butyl phthalate*<br>*di-n-nonyl phthalate*<br>*diethyl phthalate*<br>*diisobutyl phthalate*<br>*ethylene brassylate*<br>*galaxolide*<br>*n,n-diethyl-m-toluamide*<br>*tonalide*<br>1-methylnaphthalene<br>4-chlorophenyl isocyanate<br>anthracene<br>benzyl benzoate<br>butyl benzyl phthalate<br>caffeine<br>coumarin<br>d-limonene<br>di-*n*-octyl phthalate<br>exaltolide (15-pentadecanolide)<br>lilial<br>linalool<br>musk ketone<br>pyrene<br>triclosan | *amyl cinnamal*<br>*benzophenone*<br>*benzyl salicylate*<br>*bis(2-ethylhexyl)phthalate*<br>*butylated hydroxytoluene*<br>*di-n-butyl phthalate*<br>*di-n-nonyl phthalate*<br>*diethyl phthalate*<br>*diisobutyl phthalate*<br>*ethylene brassylate*<br>*galaxolide*<br>*n,n-diethyl-m-toluamide*<br>*tonalide* |

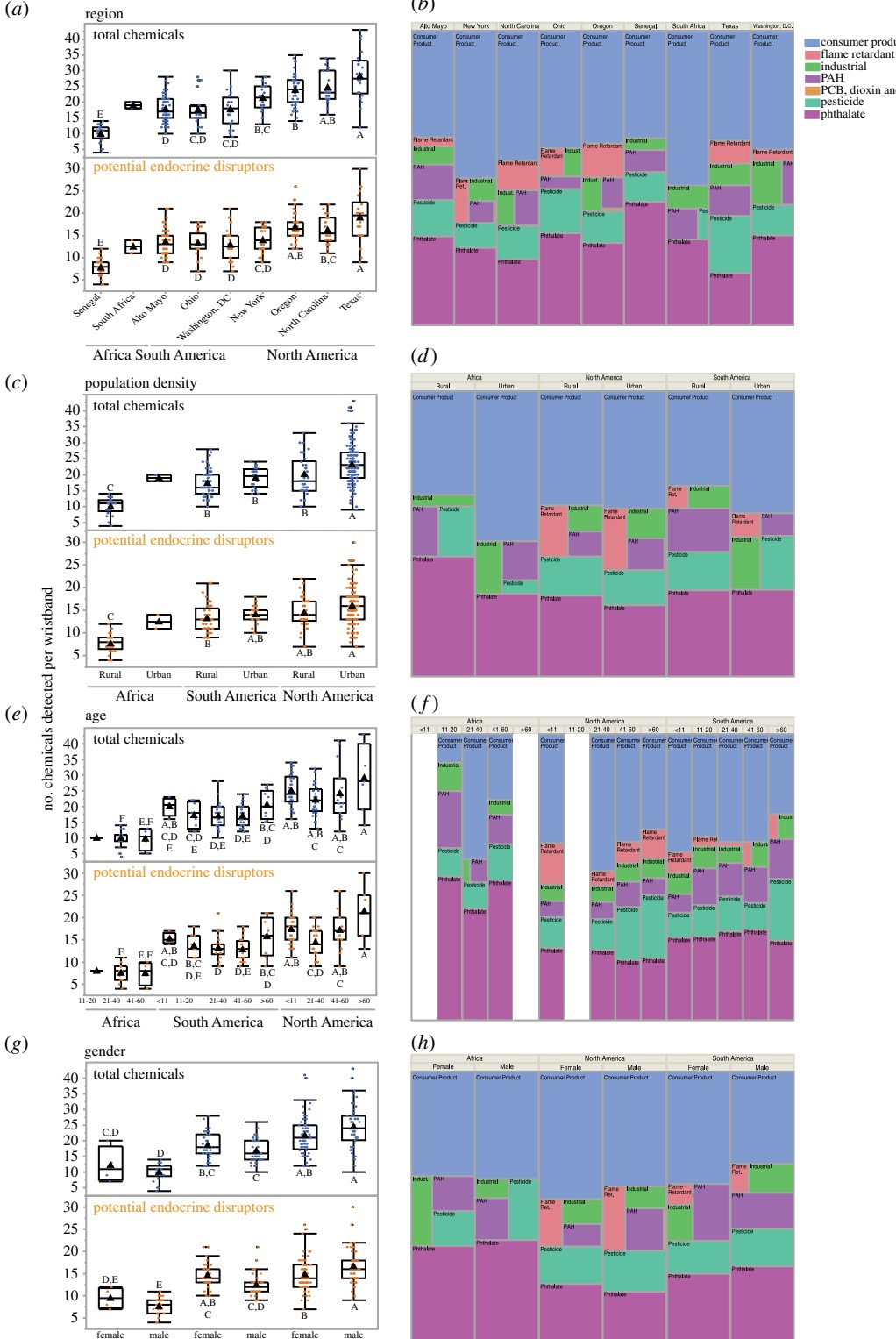

**Figure 4.** The number of chemical detections per wristband and the distribution of chemical categories are displayed for several variables: geographical region (*a,b*), population density (*c,d*), age (*e,f*) and gender (*g,h*). On the box plots, blue dots (*top*) represent the total of all chemicals detected and orange dots (*bottom*) represent the total of potential endocrine disrupting chemicals. Black triangles represent the mean number of chemical detections. For each group of boxplots, letters represent significance results; wristband group means not connected by the same letter are significantly different (Tukey – Kramer HSD, *p* < 0.05). For the tree maps, each primary chemical category (consumer product, flame retardant, industrial, PAH, PCB/dioxin/furan, pesticide and phthalate), is represented by a different colour. The size of each coloured box reflects the percentage of chemical detections for that specific category.

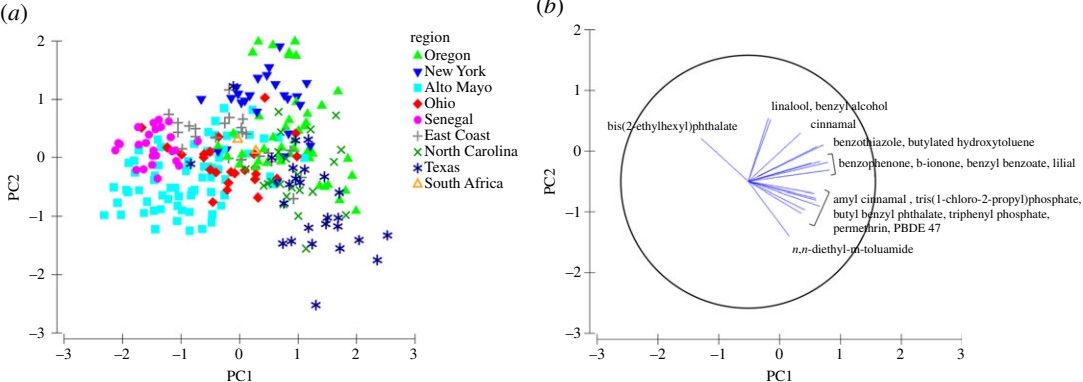

**Figure 5.** (a) Principal components analysis of PC1 and PC2 for the presence-absence chemical data from all wristbands, explaining 17% of the total variation, and (b) PC1 and PC2 explain at least 50% of the variation of each chemical vector displayed. Chemical vectors point in the direction of the increasing the density of chemical presence. Symbol shape and colour represents the region where wristbands were worn.

### 3.2.1. Region

The mean number of total chemicals detected between Africa (10.6), North America (22.5) and South America (17.8) was all significantly different from one another (Tukey–Kramer HSD, $p < 0.0001$). In figure 4a, Texas had the highest mean number of total chemicals detected (28.2) and the highest mean number of potential EDCs detected (19) compared with all other regions (electronic supplementary material, table S6). The mean number of total chemicals detected in Texas was significantly higher than the mean for Oregon, New York, Alto Mayo, Washington, DC, Ohio and Senegal (Tukey–Kramer HSD, $p = 0.01$ to $<0.0001$, figure 4a). We found similar results for potential EDCs detected (figure 4a). The mean number of total chemicals detected in Senegal (10.0) was significantly lower than the means for all other regions (Tukey–Kramer HSD, $p < 0.0001$), which was also true for the mean number of potential EDCs detected.

Texas had a high percentage of pesticide detections (19%) compared with the contribution of pesticide detections in other regions (3–15%; figure 4b). Both North Carolina and Oregon had higher percentages of flame retardants detected (10% and 12%, respectively) than the percentages in other regions (0–8%, figure 4b). Phthalates were a high percentage of Senegal's total chemical detections (42%; figure 4b). For all regions, phthalate detections were 18–42% of total chemical detections and consumer product-related chemical detections were 36–53% of total chemical detections (figure 4b).

A comparison of PC1 versus PC2 reveals similarities between wristbands worn in many different regions (figure 5). This PCA also highlights wristbands that are less similar. Many of the wristbands worn in Peru and Senegal clustered farthest to the left. In the direction of the Peru and Senegal cluster, chemical vectors suggest a lower density of certain chemical detections (such as personal care product-related chemicals like cinnamal, lilial and butylated hydroxytoluene) and higher density of bis(2-ethylhexyl)phthalate detections (presence–absence gradient shown in electronic supplementary material, figure S2). Several wristbands worn in Texas clustered as well. In the direction of the Texas cluster, chemical vectors suggest a higher density of $n,n$-diethyl-$m$-toluamide, amyl cinnamal, TCPP, butyl benzyl phthalate, TPP, permethrin and PBDE 47 detections.

### 3.2.2. Population density

The mean number of total chemicals detected between wristbands worn in rural (16.4) and urban (22.6) communities was significantly different from one another (Student's $t$-test, $p < 0.0001$). Within North America, the mean number of chemicals detected in urban wristbands (23.1) was significantly higher than the mean number of chemicals detected in rural wristbands (19.9, Tukey–Kramer HSD, $p = 0.02$, figure 4c). For South America, the mean number of chemicals detected for urban wristbands (19) was slightly higher than rural wristbands (17.3); however, this difference was not statistically significant (Tukey–Kramer HSD, $p > 0.05$; electronic supplementary material, table S7). The mean number of potential EDCs detected did not significantly differ between rural and urban groups for either North or South America (Tukey–Kramer HSD, $p > 0.05$). The mean number of chemicals detected for wristbands from rural Africa (10) was significantly lower than all other wristbands in figure 4c (17.3–23.1, Tukey–Kramer HSD, $p < 0.0001$). Although we did

not calculate statistics (because $n = 2$ in the Africa urban group, i.e. South Africa), there was a higher mean number of chemicals in the Africa urban group (19) compared with the Africa rural group (10).

Wristbands in the South America rural group had a higher percentage of PAHs (15%) and pesticides (13%) detected compared with wristbands in the South America urban group (4% and 10%, respectively; figure 4$d$). Chemical categories appear to be similar between North America rural and urban groups, with the exception of a few PCB, dioxin and furan detections in the North American urban group (0.26%). Flame retardant detections were higher contributors to total chemical detections in both North America rural and urban groups (8%) than in all other groups (0–4%). There were more pesticide detections (10%) and fewer industrial-related chemical detections (4%) contributing to the total percentage of chemical detections in the Africa rural group compared with the Africa urban group (3% and 8%, respectively).

### 3.2.3. Age

Comparing all age groups, the mean number of total chemicals detected between wristbands worn by volunteers under 11 years old (24.6) and over 60 years old (24.3) was significantly higher than the mean number for all other age groups (16.5–18.9; Tukey–Kramer HSD, $p < 0.04$ to $<0.0001$). Within each continent, the mean number of chemicals detected for each age group was not significantly different (Tukey–Kramer HSD, $p > 0.05$, figure 4$e$). Similar to other variables, age groups in Africa had significantly lower mean numbers of chemicals detected than all age groups in North America and several age groups in South America (Tukey–Kramer HSD, $p = 0.04$ to $<0.00001$; electronic supplementary material, table S8).

Flame retardants were present in all age groups in North America and South America (figure 4$f$). Flame retardant detections contributed more to the total percentage of chemical detections in wristbands worn by North Americans under age 11 (15%) compared with flame retardant detections in other age groups in North Americans (5%, 7% and 10%; figure 4$f$). Common flame retardants in these children's wristbands included TCPP, tributyl phosphate, PBDE 47, TPP and tris(2-ethylhexyl) phosphate. Flame retardants were not detected in wristbands worn by volunteers of any age in Africa.

Consumer product-related chemicals and phthalates were a high percentage of chemicals detected in all age groups for each continent (10–48% for consumer product-related chemicals, and 21–50% for phthalates; figure 4$f$). In Africa and North America, the highest percentage of consumer product-related chemicals was for volunteers between 21 and 40 years old. In North America and South America, pesticide detections contributed more to the total percentage of chemical detections in wristbands worn by volunteers over 60 compared with wristbands worn by younger volunteers (figure 4$f$).

### 3.2.4. Gender

The mean number of total chemicals detected between wristbands worn by females (20.3) and males (18.9) was not significantly different from one another (Student's $t$-test, $p = 0.14$). Additionally, within each continent, there were no significant differences found between the mean total number of chemicals detected for males and females (Tukey–Kramer HSD, $p > 0.05$, figure 4$g$; electronic supplementary material, table S9). North American males had a higher mean number of chemicals detected (24.5) than North American females (21.7), even if not a significant difference (Tukey–Kramer HSD, $p = 0.07$). South American females had a higher mean number of chemical detections (18.6) than South American males (16.8), but this was not a significant difference (Tukey–Kramer HSD, $p = 0.74$). When looking at the mean number of potential EDCs detected, there was a significantly higher mean for North America males (16.7) than North America females (14.8, Tukey–Kramer HSD, $p = 0.04$).

The chemical category profiles for each gender are similar within each continent (figure 4$h$). For North and South America, there were slightly higher consumer product-related chemical detections for females (45% and 40%) than males (40% and 33%).

## 4. Discussion

We demonstrate that wristbands are an excellent screening tool for population exposures to organic chemicals. Wristbands are lightweight, easy to transport [22,25], do not require batteries or maintenance, and offer a unique opportunity to investigate components of the personal exposome on a global scale. Our investigation revealed that chemical exposure profiles are different between individuals and we detect significant chemical detection differences based on geography, population

density, age and gender variables. To our knowledge, this is the first study to report chemical mixture profiles for individuals on three continents and for the presence–absence of over 1500 organic chemicals.

Although wristbands worn by volunteers detected different chemicals, we identified notable patterns. For example, 14 chemicals were detected in over 50% of wristbands, 13–36 chemicals were found in common between different groups of wristbands and several co-occurring pairs of chemicals were identified. Our results reveal common chemical mixtures across several communities that can be prioritized for future toxicology and epidemiology studies [4,52]. Toxicologists can investigate health effects resulting from the common mixtures we report. In addition, if there is a certain chemical of concern (e.g. 'chemical Y'), toxicologists can recreate a mixture of the common chemicals reported in the study plus 'chemical Y' to investigate potential non-additive interactions. Exposure scientists can also use these mixtures from our extensive chemical screen to begin to assess the relative concentrations of these mixtures in different populations.

When considering the estimated 140 000 chemicals synthesized worldwide since 1950 [53], research can be hindered by the possible number of chemical mixtures that need to be investigated. Yet, it is unlikely that people are exposed to a mixture of all 140 000 chemicals. For example, we screened for 1530 priority chemicals, but only detected 191 different chemicals. It is likely that not all chemicals reach bioactive sites because physical-chemical properties can limit chemical bioavailability. Previous research has demonstrated that passive samplers, including wristbands, reflect the bioavailable fraction of chemicals [23,29,35,40]. Thus, investigating all factorial combinations of chemical mixtures could be simplified by focusing on patterns that are detected in multiple populations using passive sampling technology.

There are several approaches for studying chemical mixtures, including the 'sufficiently similar' mixture approach proposed by the U.S. Environmental Protection Agency [54]. In an attempt to balance practicality and limited scope issues present with other mixture assessment methodologies, like whole mixture and component-based approaches [55], a sufficiently similar mixture is where proportions between chemical components match the real-world mixture [54]. Such methodologies could be applied moving forward with the common chemicals reported in this study. For example, Geier *et al*. applied this approach and constructed a fixed-ratio, environmentally relevant PAH mixture with the top 10 most prevalent PAHs from a Superfund site and used zebrafish to assess developmental and neurotoxicological hazards [55]. Researchers can use wristband data to prioritize and create sufficiently similar mixtures for investigating the effect of chemical exposures on human health outcomes.

To our knowledge, this is the first study to screen for personal exposure to 432 potential EDCs in samplers worn by volunteers on three continents. A 2015 review on EDCs summarizes different ways to classify EDCs, such as (i) those that occur naturally versus synthesized [56] and (ii) those with different origins, including natural and artificial hormones, drugs with hormonal side effects, industrial and household chemicals and side products of industrial and household processes [13,57]. Here, we detected 13 potential EDCs in over 50% of wristbands. Since these 13 potential EDCs are common and bioavailable, this is a potential new EDC category to prioritize in future studies. The baseline EDC exposure data we report may help researchers link emerging health issues with EDC exposure.

## 4.1. Chemical detection comparisons between geographic and demographic variables

Because phthalates and consumer product-related chemicals make up a large percentage of personal chemical exposure, regardless of a person's age, gender or location, these chemical categories may be a high priority for future toxicology and epidemiology studies. Phthalates are plasticizers, some of which are known EDCs (54). Human exposure to certain phthalates has been associated with adverse male reproductive outcomes and impaired behavioural development [58–61]. Owing to health concerns, the U.S. federal government passed legislature in 2008 banning the use of di-*n*-butyl phthalate, bis(2-ethylhexyl)phthalate and butyl benzyl phthalate in concentrations of more than 0.1% in children toys and certain child care articles [62], which we detected in this study in 92%, 84% and 66% of wristbands, respectively. Additional wristband studies can help assess phthalate exposure temporal trends.

Our results also indicate that personal PCB exposure is not common for inhalation and dermal routes at concentrations above our detection limits. Notably, there are 260 PCBs/dioxins/furans in our analytical method, but PCBs were only detected eight times in wristbands worn in either New York or Oregon. PCB exposure has been linked to many health issues such as cancer [63] and immune system issues [64], leading to the phase-out of these chemicals under the Toxic Substances Control Act in the U.S. and Stockholm Convention [65].

### 4.1.1. Region

Although we found common chemical exposures, there are also distinct differences between wristbands worn in different regions. For example, wristbands worn by people near Houston, Texas within a month after Hurricane Harvey landfall (i) had a significantly higher mean number of chemical detections than several other geographical regions, (ii) had a relatively higher number of pesticide detections than other regions and (iii) clustered during PCA. In communities affected by disasters, personal chemical exposure is probably unique and wristbands can assess exposure during critical time windows.

The significantly lower mean number of chemical detections in Senegal compared with other regions, and cluster of Senegal wristbands in PCA, might be due to differences in behaviours and built environment (human-made surroundings) compared with other regions. The absence of flame retardants in both Senegal and South Africa wristbands may reflect a difference in flammability protection standards [66], housing materials and/or furniture used in certain Africa communities compared with other communities in North and South America. A 2016 review on exposure studies in Africa states that PBDE flame retardants have been found in dust, soil, water and human breast milk [67]. In Tanzania, PBDEs in breast milk were found in higher concentrations than Asia and Europe, but it is noted that PBDEs were inconsistently detected in samples from Africa and few samples have been reported from West Africa [67].

### 4.1.2. Age

North Americans in this study, especially those under 11 years old, have a higher percentage of flame retardant detections compared with all other groups, highlighting priorities for future studies. PBDE flame retardants are known to be neurotoxic, and children with higher exposure to PBDEs have been associated with a greater risk of neurological issues such as negative social behaviours, reduced verbal comprehension and working memory, and autistic-like behaviours [68,69]. Many PBDEs are no longer used because of their persistence and concerns about their effect on children, leading to greater use of OPFRs. However, there is evidence that OPFRs are also neurotoxic, potentially using the same mechanisms as organophosphate pesticides [70].

Because of a higher contribution of consumer product-related chemical detections, North American participants between the ages of 21 and 40 may have used more personal care products than other age groups in this study. In addition, individuals over the age of 40 in this study may have handled more pesticides to control pests around the home or for agricultural purposes. Pesticide exposure can result in a variety of adverse health effects [71].

### 4.1.3. Population density

Prioritization of exposure assessment to certain chemical classes might be different between rural and urban communities. For example, PAH detections were higher in South America rural than South America urban. Differences in heating sources and other behaviours related to cooking, burning, smoking and vehicle exhaust might contribute to those differences. Additionally, our definitions of rural and urban may have influenced our results. For example, we did not have information available in this study if volunteers were living in a rural area but spending large amounts of time in urban locations.

### 4.1.4. Gender

Males and females had similar mean numbers of chemicals detected within continents, which may indicate similar behaviours and built environments. Future studies could focus on male EDC exposure because we observe a significant increase in the mean number of potential EDC detections in males compared with females in North America. Significant knowledge gaps exist, but researchers have hypothesized that EDC exposure contributes to developmental genital anomalies and low semen quality [72]. As of 2013, significant percentages of young men (up to 40% in some countries) were reported to have low semen quality [72], further warranting additional research on male EDC exposure.

## 4.2. Additional considerations

While this is the first study to screen for the presence of 1530 organic chemicals in wristbands worn on three continents, there are limitations worth noting. We relied on a convenience sample of volunteers and did not randomly recruit participants in this study. Therefore, the chemical exposures we report may not

be representative of all chemical exposures in the 14 communities included. Even so, these data are the first exploration of organic chemical exposures detected by wristbands across diverse communities, helping inform future research priorities. Additionally, our results do not reflect exposure from particulate-bound chemicals or from ingestion because we use wristbands to sample organic chemicals in the gaseous phase important to inhalation and dermal exposure routes. In the future, as the number of studies using wristbands increase, it would be beneficial to standardize questionnaires and IRBs so researchers can build a robust database to explore personal chemical exposure.

This is an exploratory, retrospective study, so wristband deployment length varied depending on the specific project. In this study, we did not detect a difference in the number of chemicals detected based on how long a participant wore a wristband (electronic supplementary material, figure S3).

We have communicated chemical exposure data to most volunteers included in this study. There are many considerations when returning chemical results, with special care not to cause harm but rather to increase knowledge about chemical exposure. Studies have shown that participants report benefits from receiving their results even if exposure limits and health effects are uncertain [73–75]. Returning results can give participants insight into the study they participated in and offer them the opportunity to make their own decisions about their chemical exposures [73,76,77]. Brody et al. state that 'for example, participants may choose to reduce exposures as a precaution or to become engaged in public discourse about chemical use and regulation' [76]. We will continue to report chemical results to participants, and we anticipate that we will incorporate the results from this study into participant reports.

## 5. Conclusion

Wristbands sampled personal exposure to a wide range of consumer product-related chemicals, flame retardants, industrial-related chemicals, PAHs, PCBs/dioxins/furans, pesticides and phthalates. Owing to the innovation and applicability of wristbands in exposure science studies over the past few years, we were able to compare chemical detection data between 14 different communities on three continents, resulting in four primary conclusions:

(1) Not all synthesized chemicals are in the personal environment and bioavailable. Out of the 1530 chemicals in our chemical method, we detected 191 unique chemicals.
(2) Personal chemical exposure varies by individual. No two wristbands had identical chemical detection profiles.
(3) Patterns in personal chemical exposure emerged, including the detection of 14 chemicals in over 50% of wristbands, revealing common mixtures that can be used in future toxicology research on chemical mixtures. These common chemicals are primarily potential EDCs (93%).
(4) Geographical and demographic variables highlight priority chemical categories for future studies, such as flame retardant exposure in North American children and chemical exposure in communities affected by natural disasters (e.g. people in Houston, Texas after Hurricane Harvey-related flooding).

Gathering personal exposure data with wristbands can be valuable for informing how organic chemical mixtures, especially EDCs, influence health.

Research ethics. All research activities were granted prior approval by Institutional Review Boards (IRBs) including Oregon State University IRBs nos. 5338, 5736, 6479, 8058 and 8146; Columbia University IRB no. AAAK6753; Medical College of Wisconsin IRB no. 7221; University of Cincinnati IRB no. 2013-4095; Pacific Northwest National Laboratory IRB no. 8058 and Wake Forest School of Medicine IRB no. IRB00037775. All volunteers provided written or verbal consent prior to data collection, as per IRB.
Data accessibility. The dataset supporting this article has been uploaded as the electronic supplementary material. The following information is also available in the electronic supplementary material: chemical lists corresponding to the Venn diagrams, all Tukey–Kramer HSD results, network analysis figure visualizing pairs of chemicals detected together greater than or equal to 20 times, demonstration of presence–absence density in PCA and number of chemical detections compared with wristband deployment time.
Authors' contributions. H.M.D. performed chemical and statistical analyses, created tables and figures, and wrote the manuscript. A.J.B. and P.N. coordinated wristband deployment and processing in Peru. L.B.P. and E.H. facilitated the collection of the Ohio wristbands. M.L.H. and P.J. facilitated and helped with the collection of wristbands in Senegal. C.M.P. and P.J.L. participated in North Carolina wristband collection and analysis. M.L.K. was part of the team that collected samples in Oregon. D.R. assisted with data collection in Eugene, Ohio and Texas. L.K. assisted with data collection in Ohio and Oregon. L.G.T. and P.H. assisted with Texas wristband deployment and analysis.

G.A., M.B., W.H., J.P. and C.W. assisted with organizing the Texas wristband study. K.M.W., L.K. and J.H. conceived and designed the New York and Eugene wristband studies. M.B. helped with data management and created the network analysis visualization. B.S. provided statistical advice, generated the data for the Venn diagram tables and performed principal component analysis. G.L.P. and R.P.S. helped with all wristband preparation and analysis. K.A.A. conceived and designed this study and oversaw all chemical analyses. All authors reviewed and edited the manuscript, and gave their final approval for publication.

Competing interests. K.A.A. and D.R., authors of this research, disclose a financial interest in MyExposome, Inc., which is marketing products related to the research being reported. The terms of this arrangement have been reviewed and approved by OSU in accordance with its policy on research conflicts of interest. The authors have no other disclosures.

Funding. Research reported in this publication was supported by the National Institute of Environmental Health Sciences (NIEHS) under award numbers P42 ES012016465, P20 ES000210, R21 ES020120, R33 ES024718, R24 TW009550, P30 ES000210, P30 ES006096 and R01 ES008739. H.M.D. was supported in part by NIEHS Fellowship T32 ES007060 and ARCS Foundation® Oregon. Pacific Northwest National Laboratory is a multi-program national laboratory operated by Battelle for the U.S. Department of Energy under Contract DE-AC05-76RL01830.

Acknowledgements. We wish to thank the volunteer participants in this study. From Oregon State University's Food Safety and Environmental Stewardship Program, we thank Glenn Wilson, Steven O'Connell, Christine Ghetu, Rachel Liu-May, Clarisa Caballero-Ignacio, Ian Moran, Jessica Scotten, Jorge Padilla, Melissa McCartney, Josh Willmarth and Amber Barnard. We acknowledge Carey Donald for her work with the wristbands from Senegal. We thank Sara A. Quandt and Thomas A. Arcury from Wake Forest for their help with North Carolina and South Africa wristbands, and Hanna-Andrea Rother from the University of Cape Town for help with the collection of the South Africa wristbands. We acknowledge Darrell Holmes and Lehyla Calero for their help with the collection of wristbands in New York. We acknowledge Jackie Young, Robin Fuchs-Young, Madison Spier, Oluwatosin Bewaji, Gustavo Elizondo and the staff of Texas Health and Environment Alliance, Inc. for organizational support in Houston, Texas. For the wristband project in Ohio, we thank Paul Feezel, Kevin Hobbie, Sarah Elam, Delores Silverthorn, David Brown, Jody Alden and Pierce Kuhnell. We also acknowledge Lisa Arkin for her assistance with the Eugene project.

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
