## [Reviewer comments · Royal Society Open Science]

Review History

RSOS-181836.R0 (Original submission)

Review form: Reviewer 1

Is the manuscript scientifically sound in its present form?

Yes

Are the interpretations and conclusions justified by the results?

Yes

Is the language acceptable?

Yes

Is it clear how to access all supporting data?

Yes

Do you have any ethical concerns with this paper?

No

Have you any concerns about statistical analyses in this paper?

I do not feel qualified to assess the statistics

Recommendation?

Accept with minor revision (please list in comments)

Comments to the Author(s)

This article can be accepted after the following minor revisions:

1. Please avoid first person terms such as 'we', 'our', etc. Rephrase those sentences with person passive.
2. Experimental details for the methods can be briefed before providing the references simply. E.g. page 4 - preparation & deployment of wristband; page 5 - extraction of wristband, SPE extraction, GC & MSD method parameters, AMDIS processes, etc. and page 7 - for more QC details and results on specific projects.
3. Details on blank wristbands (or the controls) is not clear and sufficient. Should explain how this is maintained and compared with actual wristbands under exposures.
4. For such wider spectrum of chemicals what were the internal standards employed for GC-MS technique? Based on which USEPA method (or other equivalent) the analytical procedures were established?
5. How the LODs and LOQs were determined and, should provide those details in supplementary data.
6. Why the variation for the number of volunteers and number of wristbands?
7. While conducting this study were the communities asked to adhere to their routine tasks as usual including the application of cosmetics, toiletries, washing/cleaning chemicals, etc? How it has been practiced and what was the time period?
8. Any variation observed depending on the work/job sites of the studied persons?

Review form: Reviewer 2 (Tawfik Saleh)

Is the manuscript scientifically sound in its present form?

Yes

Are the interpretations and conclusions justified by the results?

Yes

Is the language acceptable?

Yes

Is it clear how to access all supporting data?

Yes

Do you have any ethical concerns with this paper?

No

Have you any concerns about statistical analyses in this paper?

No

Recommendation?

Major revision is needed (please make suggestions in comments)

Comments to the Author(s)

The work reported in this manuscript is interesting. However, it needs revisions and improvements before the acceptance. Some comments are:

1. Title should be revised to be precise and reflecting the contents;
2. No need to mention the abbreviation in the abstract unless it is reused.
3. Abstract is too short. Abstract should be rewritten to summarize the work; the abstract should state briefly the purpose of the research, the principal results and major conclusions. An abstract is often presented separately from the article, so it must be able to stand alone.
4. In the abstract; "highest mean number of chemical detections....." pls add quantitative data
5. Define the terms and abbreviations when used.
6. The introduction should be clarified in term of uniqueness and advantage what is the novelty of this work over the previous related work. There are many long sentences should be refined.
7. Page 2; sec Study Participants and Design: was introduced without enough details on the steps and conditions, etc.
8. Page 7; Network Analysis. Please improve the presentation of this section
9. Under introduction, please after "to specific flame retardants has been associated.." please add Journal of Chemical Health and Safety 18 (2), 3-8; Journal of Nano Education 4 (1-2), 1-7
10. Please revise the conclusion
11. English must be improved.
12. Add experimental conditions to captions of each figure.

I WOULD LIKE TO SEE THE REVISED MANUSCRIPT.

Decision letter (RSOS-181836.R0)

27-Nov-2018

Dear Dr Anderson:

Title: Discovery of common chemical exposures across three continents using silicone wristbands
Manuscript ID: RSOS-181836

The editor assigned to your manuscript has now received comments from reviewers. We would like you to revise your paper in accordance with the referee and Subject Editor suggestions which can be found below (not including confidential reports to the Editor). Please note this decision does not guarantee eventual acceptance.

Please submit your revised paper before 20-Dec-2018. Please note that the revision deadline will expire at 00.00am on this date. If we do not hear from you within this time then it will be assumed that the paper has been withdrawn. In exceptional circumstances, extensions may be possible if agreed with the Editorial Office in advance. We do not allow multiple rounds of revision so we urge you to make every effort to fully address all of the comments at this stage. If deemed necessary by the Editors, your manuscript will be sent back to one or more of the original reviewers for assessment. If the original reviewers are not available we may invite new reviewers.

To revise your manuscript, log into <http://mc.manuscriptcentral.com/rsos> and enter your

Author Centre, where you will find your manuscript title listed under "Manuscripts with Decisions." Under "Actions," click on "Create a Revision." Your manuscript number has been appended to denote a revision. Revise your manuscript and upload a new version through your Author Centre.

RSC Associate Editor:
Comments to the Author:
(There are no comments.)

RSC Subject Editor:
Comments to the Author:
(There are no comments.)

Reviewers' Comments to Author:
Reviewer: 1

Comments to the Author(s)

This article can be accepted after the following minor revisions:

1. Please avoid first person terms such as 'we', 'our', etc. Rephrase those sentences with person passive.
2. Experimental details for the methods can be briefed before providing the references simply. E.g. page 4 - preparation & deployment of wristband; page 5 - extraction of wristband, SPE extraction, GC & MSD method parameters, AMDIS processes, etc. and page 7 - for more QC details and results on specific projects.
3. Details on blank wristbands (or the controls) is not clear and sufficient. Should explain how this is maintained and compared with actual wristbands under exposures.

4. For such wider spectrum of chemicals what were the internal standards employed for GC-MS technique? Based on which USEPA method (or other equivalent) the analytical procedures were established?
5. How the LODs and LOQs were determined and, should provide those details in supplementary data.
6. Why the variation for the number of volunteers and number of wristbands?
7. While conducting this study were the communities asked to adhere to their routine tasks as usual including the application of cosmetics, toiletries, washing/cleaning chemicals, etc? How it has been practiced and what was the time period?
8. Any variation observed depending on the work/job sites of the studied persons?

Reviewer: 2

Comments to the Author(s)

The work reported in this manuscript is interesting. However, it needs revisions and improvements before the acceptance. Some comments are:

1. Title should be revised to be precise and reflecting the contents;
 2. No need to mention the abbreviation in the abstract unless it is reused.
 3. Abstract is too short. Abstract should be rewritten to summarize the work; the abstract should state briefly the purpose of the research, the principal results and major conclusions. An abstract is often presented separately from the article, so it must be able to stand alone.
 4. In the abstract; "highest mean number of chemical detections....." pls add quantitative data
 5. Define the terms and abbreviations when used.
 6. The introduction should be clarified in term of uniqueness and advantage what is the novelty of this work over the previous related work. There are many long sentences should be refined.
 7. Page 2; sec Study Participants and Design: was introduced without enough details on the steps and conditions, etc.
 8. Page 7; Network Analysis. Please improve the presentation of this section
 9. Under introduction, please after "to specific flame retardants has been associated.." please add Journal of Chemical Health and Safety 18 (2), 3-8; Journal of Nano Education 4 (1-2), 1-7
 10. Please revise the conclusion
 11. English must be improved .
 12. Add experimental conditions to captions of each figure.
- I WOULD LIKE TO SEE THE REVISED MANUSCRIPT.

Author's Response to Decision Letter for (RSOS-181836.R0)

See Appendix A.

RSOS-181836.R1 (Revision)

Review form: Reviewer 1

Is the manuscript scientifically sound in its present form?

Yes

Are the interpretations and conclusions justified by the results?

Yes

Is the language acceptable?

Yes

Is it clear how to access all supporting data?

Yes

Do you have any ethical concerns with this paper?

No

Have you any concerns about statistical analyses in this paper?

No

Recommendation?

Accept as is

Comments to the Author(s)

After the amendments based on my suggestions, the article can be recommended for publication without any further revision.

Decision letter (RSOS-181836.R1)

14-Jan-2019

Dear Dr Anderson:

Title: Discovery of common chemical exposures across three continents using silicone wristbands
Manuscript ID: RSOS-181836.R1

It is a pleasure to accept your manuscript in its current form for publication in Royal Society Open Science. The chemistry content of Royal Society Open Science is published in collaboration with the Royal Society of Chemistry.

The comments of the reviewer(s) who reviewed your manuscript are included at the end of this email. I apologise that this took longer than usual.

RSC Associate Editor:
Comments to the Author:
(There are no comments.)

RSC Subject Editor:
Comments to the Author:
(There are no comments.)

Reviewer(s)' Comments to Author:
Reviewer: 1

Comments to the Author(s)
After the amendments based on my suggestions, the article can be recommended for publication without any further revision.

Appendix A

Response to Referees [RSOS-181836.R1]

December 1, 2018

Authors: We would like to thank the reviewers for their careful consideration. We have done our best to address each concern directly and where appropriate within the manuscript. We highlighted our responses in blue text below.

All page and line number references refer to the tracked changes version of the manuscript.

Referee Comments:

Reviewer: 1

Comments to the Author(s)

This article can be accepted after the following minor revisions:

1. Please avoid first person terms such as 'we', 'our', etc. Rephrase those sentences with person passive.

We double-checked the author guidelines for RSOS and there is no specification for either passive or active voice. The top three most cited RSOS papers all use active voice (e.g. use first person terms including “we”). We chose to write in active voice to make the manuscript more concise and increase readability.

2. Experimental details for the methods can be briefed before providing the references simply. E.g. page 4 - preparation & deployment of wristband; page 5 - extraction of wristband, SPE extraction, GC & MSD method parameters, AMDIS processes, etc. and page 7 - for more QC details and results on specific projects.

We thank the reviewer for this comment, and we added additional specifics throughout the methods section (page 5 lines 166-173, 176-178, 184-187, 192-193, 195, 204-207; page 6 lines 231-239). We describe every laboratory process alongside appropriate references. We also added a table to the supplemental material which includes the gas chromatography-mass spectrometry parameters (now Table S1).

3. Details on blank wristbands (or the controls) is not clear and sufficient. Should explain how this is maintained and compared with actual wristbands under exposures.

We appreciate this comment. While this paper is not a review of quality control results from all previously published studies, we edited and added additional details pertaining to blank wristbands within the section titled “Quality Control Summary” on page 6 (lines 231-239). We also relocated information pertaining to blank wristbands from the wristband conditioning process under the “Preparation & Deployment” section to make this clearer.

4. For such wider spectrum of chemicals what were the internal standards employed for GC-MS technique? Based on which USEPA method (or other equivalent) the analytical procedures were established?

We thank the reader for the opportunity to clarify. There is no EPA method or equivalent with >1,500 organic chemicals. This novel method is reported in Bergmann et al. 2018 and is based

Response to Referees [RSOS-181836.R1]

on a target screening method. Target screening methods are frequently performed with GC or liquid chromatography. With target screening, the full scan ion profile is captured and deconvolution software is used to extract specific signals from complex chromatograms. For this work, we used the Automated Mass Spectral Deconvolution and Identification System (AMDIS, from the **National Institute of Standards and Technology**) paired with commercially available deconvolution reporting software from **Agilent** (page 5, lines 194-197). The method is retention time locked to an internal standard, chlorpyrifos (page 5, line 194), and additional internal standards are not needed for this presence-absence method. We use calibration verifications (CVs) to monitor instrument conditions (page 6, lines 223-227), which include standards that span the range of physical chemical properties in the method. For the quantification version of this method (separate from this manuscript and fully described in Bergmann et al. 2018), additional standards and chemometrics are used to predict instrument response. For the presence-absence target screen in this manuscript, it is suitable to present target screen software (page 5), chemical libraries (page 5), CV data quality objectives (page 6), and GC-MS operating parameters (now added as Table S1 in response to comments).

5. How the LODs and LOQs were determined and, should provide those details in supplementary data.

Bergmann et al. used a standard average response variation method to estimate instrument LOQs for all chemicals in the method. LOQs range from 40 to 500 pg/uL depending on the chemical (page 5, lines 201-203; Bergmann et al. 2018). Because LOQs are reported in a previously published manuscript for all 1,530 chemicals and we are presenting presence-absence data, it is appropriate to present the range of the LOQs for this method and reference the original manuscript. The data quality objectives we used to evaluate the presence/absence of a chemical protects against false positives and we have now added this to the manuscript (page 5, lines 204-205).

6. Why the variation for the number of volunteers and number of wristbands?

This was an exploratory, retrospective study (page 4, lines 149-150; page 12, line 558) so the number of volunteers and number of wristbands was opportunistic and based on existing wristband projects. These projects had different funding sources, collaborators, and original study questions – resulting in variation in the number of volunteers and number of wristbands per geographic community. We discuss the limitation of a convenience sample of volunteers on page 12, lines 548-553. However, this is the “first exploration of organic chemical exposures detected by wristbands across diverse communities, helping inform future research priorities” (page 12).

7. While conducting this study were the communities asked to adhere to their routine tasks as usual including the application of cosmetics, toiletries, washing/cleaning chemicals, etc?
How it has been practiced and what was the time period?

All participants were asked to maintain their normal daily activities and not change their behavior for the study. We added this information to page 4 (lines 154-155). Study time period and other community specifics are located in Table 1 and supplemental data set.

8. Any variation observed depending on the work/job sites of the studied persons?

Previous wristbands have reported chemical exposure variation in relation to work/job sites (Bergmann et al. 2017, O'Connell et al. 2014). However, we did not include work/job variables in

Response to Referees [RSOS-181836.R1]

this manuscript because we did not have information on these variables for a majority of wristbands.

Reviewer: 2

Comments to the Author(s)

The work reported in this manuscript is interesting. However, it needs revisions and improvements before the acceptance. Some comments are:

1. Title should be revised to be precise and reflecting the contents;

We agree the title should be a short description of the research and we include words to help readers discover this paper. After considering both reviewers' comments, we decided to keep our title as is (Discovery of common chemical exposures across three continents using silicone wristbands).

2. No need to mention the abbreviation in the abstract unless it is reused.

The only abbreviation in the abstract is "U.S.", which we have now changed to "United States" on line 45. This abbreviation is reused in the abstract.

3. Abstract is too short. Abstract should be rewritten to summarize the work; the abstract should state briefly the purpose of the research, the principal results and major conclusions. An abstract is often presented separately from the article, so it must be able to stand alone.

We agree the abstract should be standalone. We summarize the purpose, results, and major conclusions within the abstract. Per RSOS guidelines, the abstract must be concise and under 200 words, and we were at 198 words upon submission. We did add to the word count to address comment 2 and comment 4.

4. In the abstract; "highest mean number of chemical detections....." pls add quantitative data

We thank the reviewer for this comment and added in quantitative results to this sentence.

5. Define the terms and abbreviations when used.

Per RSOS guidelines, we defined and wrote out each abbreviation in full upon first use.

6. The introduction should be clarified in term of uniqueness and advantage what is the novelty of this work over the previous related work. There are many long sentences should be refined.

We thank the reviewer for this comment. Within the introduction, we highlight the novelty of this work by describing wristbands as an "unprecedented measurement platform", the need for "simple personal monitoring methods", and the generation of data to "inform future toxicology and epidemiology research." We made edits to the sentence structure where applicable.

7. Page 2; sec Study Participants and Design: was introduced without enough details on the steps and conditions, etc.

Response to Referees [RSOS-181836.R1]

In response to both reviewers, we added additional details about participant behavior during the study on lines 154-155 of page 4. Table 1 provides the following steps and conditions of the studies including: community, region, number of volunteers, number of wristbands, time of study, related reference, and number of wristbands/percentages for population density, age, and gender variables.

8. Page 7; Network Analysis. Please improve the presentation of this section

We improved the presentation of the network analysis figure (Figure S1). We have moved the location of the chemical names to improve figure interpretation.

9. Under introduction, please after “to specific flame retardants has been associated..” please add Journal of Chemical Health and Safety 18 (2), 3-8; Journal of Nano Education 4 (1-2), 1-7

We thank the reviewer for suggesting additional references here. We have added two additional citations (Kim et al. 2014, van der Veen et al. 2012) after the phrase “to specific flame retardants has been associated”, which address health effects associated with flame retardant exposure.

We have investigated the two references suggested by the reviewer, which are titled:

- “Testing the effectiveness of visual aids in chemical safety training” by Tawfik A. Saleh
- “A strategy for integrating basic concepts of nanotechnology to enhance undergraduate nano-education: Statistical evaluation of an application study” by Tawfik A. Saleh

These references do not mention flame retardants or relate to our manuscript’s subject matter so we did not reference them in our manuscript.

10. Please revise the conclusion

The four primary conclusions we list are strongly supported by the data presented in our results/discussion. Without specifics on how to revise and since reviewer 1 did not comment on the conclusion, we did not make changes to the conclusion.

11. English must be improved .

Thank you for this comment. The authors have done another round of edits.

12. Add experimental conditions to captions of each figure.

We have revisited RSOS author guidelines which state that figure captions should be brief and informative. We do not want to repeat information previously provided in the methods section. We reviewed the top cited RSOS papers and these papers do not list experimental conditions within the figure captions.

I WOULD LIKE TO SEE THE REVISED MANUSCRIPT.